# The Role of Systems Biologic Approach in Cell Signaling and Drug Development Responses—A Mini Review

**DOI:** 10.3390/medsci6020043

**Published:** 2018-05-30

**Authors:** Vrushali Abhyankar, Paul Bland, Gabriela Fernandes

**Affiliations:** 1Department of Periodontics, University of Tennessee Health Science Center, Memphis, TN 38163, USA; pbland@uthsc.edu; 2Department of Oral Biology, School of Dental Medicine, SUNY Buffalo, New York, NY 14215, USA; gfernand@buffalo.edu

**Keywords:** systems biology, chemokines, cytokines

## Abstract

The immune system is an integral aspect of the human defense system and is primarily responsible for and involved in the communication between the immune cells. It also plays an important role in the protection of the organism from foreign invaders. Recent studies in the literature have described its role in the process of hematopoiesis, lymphocyte recruitment, T cell subset differentiation and inflammation. However, the specific molecular mechanisms underlying these observations remain elusive, impeding the elaborate manipulation of cytokine sequential delivery in tissue repair. Previously, the discovery of new drugs and systems biology went hand in hand; although Systems biology as a term has only originated in the last century. Various new chemicals were tested on the human body, and studied through observation. Animal models replaced humans for initial trials, but the interactions, response, dose and effect between animals and humans could not be directly correlated. Therefore, there is a need to form disease models outside of human subjects to check the effectiveness and response of the newer natural or synthetic chemicals. These emulate human disease conditions wherein the behavior of the chemicals would be similar in the disease model and humans.

## 1. Introduction

The information available for the various immune system pathways has been developed through the reductionist approach where-in different pathways, cells and their interactions have been individually studied [1,2,3,4,5]. The sheer magnitude and scope of the information and understanding the relation and dependence of various factors are challenging. Various attempts have been made to integrate the information and form a predictive inflammatory model to define the precise chemokine and cytokine response to the various pro-inflammatory agonists [5,6,7,8]. Unearthing the intricate aspects of the immune responses in the disease and the interaction of the gene pool and its phenotypical expression creates a global perspective of the systems biologic approach. This helps to chart out the holistic relationships between different data sets from a large amount of information [9].

Systems biology in its simplest form is the study of biological systems to understand their complexities [10]. It involves the integration of several disciplines such as mathematics, engineering, computer science, physics, chemistry and biology to comprehend this process [10]. It also combines experimental and computational research to understand network behavior between gene protein and informational pathways, requiring an experimental basis and mathematical modeling [11]. Such models characterize the behavior of the biologic network, which goes beyond the description of existing knowledge by assigning relevance to a particular component or mechanism [11]. Furthermore, systems biology also implements a holistic approach rather than the traditional reductionist approach and tries to understand the role of cells and tissues functioning as a whole rather than individual components. Thus, animal targets led to the development of tissue level screens, simple cell-based pathway screens and the most modern ultra-high throughput screens, delineating thousands of compounds in a very short time [10,11,12].

Mining huge amounts of data in immunology to comprehend a broad picture is an extremely difficult task. Various analytical strategies have been proposed to simplify this task to delineate responses of diverse cells to any stimulation for, e.g., a pro-inflammatory agonist [11]. In immunological studies, a network analysis approach can be used to understand the various constituents and the pathways through which these constituents interact. This course is mainly used to identify the main components that regulate transcriptional programs controlling the development of immune cells, the innate immune signaling, phenotyping of immune cells through transcript profiling technologies and differentiation of pathways regulated at transcriptional level [11]. These relationships are studied as a network or graph with nodes and edges. The nodes are the various elements and the edges are the relationships between these constituents, the information of which is achieved by excavating through the large amount of available data. However, the disadvantage of these network graphs is that they may be too complicated to decipher with the large number of pathways involved. Also, there are certain knowledge gaps of many pathways where the diversity of data and abstractions must be accounted for. If all the available data of various pathways are collated, e.g., the reactive oxygen species (ROS) a better understanding of the interactions is possible. The ROS pathway is turned on in different cell lines and different results are observed which shows that these pathways show both common and distinct effects. Demir et al. attempted to develop formal models of signaling pathways with seemingly partial or incomplete data, in their work PATIKA, (Pathway Analysis Tool for Integration and Knowledge Acquisition) where instead of a node signifying a biochemical entity, it represented an interaction or a partial pathway, which simplifies the model, represented as an acyclic graph [13]. However, multilevel abstraction cannot be simplified by this method as signaling events are only interactions and not a flow chart. Fukuda et al.’s decomposition tree and interaction graphs support both multilevel abstractions as well as interactions between the nodes [14].

Computational approaches help systematically analyze this large-scale network data [15,16,17]. The data can be made more manageable by grouping genes with similar phenotypically characteristics from a large-scale data set [15]. The modular repertoire analysis is typically a functional network derived from curated knowledge bases. [18] It involves identifying an informative feature or characteristic, reducing its dimension to a more manageable level and functionally interpreting it by all the bio-informative tools at hand [19]. Development of modular repertoire takes place in the following manner. First, transcriptome datasets are clustered together depending on their pattern and expression similarities, then sub networks called modules are formed, which identifies the modular repertoire. This in turn captures relationships between these modules and perturbations in the pathways [20]. Finally, functional characterization of these repertories helps tease apart the changes that can occur due to the transcriptional changes.

However, this system is not without its disadvantages. Since there is grouping involved, variability of among genes can go unnoticed, some genes can be missed, and heterogeneity among various genes also needs to be considered if it is to be unmasked. However, advantages of modular repertoire far outweigh the flaws. They are like frameworks, where individual genes are analyzed against them and color-coded to signify an increase or decrease in transcript abundance. Perturbations of transcriptomes can be used for assessment of immunological changes. This is an easily accessible, streamlined, cost effective way of validating the pre-existing data set.

## 2. Role of Systems Biology in Drug Discovery and Development

The drug development industry, with its extremely high failure rate, has taken keen interest in the systems biology approach [21] (see Figure 1). Presently, the development costs of drugs are high, the input with respect to time spent is significantly greater, and success depends on working with limited knowledge of biologic and physiologic processes of diseases itself. Specific known signaling pathways are usually targeted to reduce the failure risk. The challenge lies in modeling the disease outside of the human body to test the different chemicals. The human physiology, which already exists, must be represented accurately from the vast disintegrated sources of knowledge [21]. Other industries, e.g., computers run massive simulations before the product releases in the market, controlling the failure rate. However, in the drug industry, decisions about drugs are based on cell lines which are not necessarily an accurate representation of the disease [21].

Some methods to reduce the high-failure rate would be to target known signaling pathways [22], especially pathways that are proven to generate results. By formulating newer drugs to act on these known pathways or targets, it significantly reduces the risk of failure [22,23,24]. This transfers the risk to start-up companies that are financed by either bigger pharmaceutical companies or IPOs and is a well-known strategy in other industries that can be applied to drug innovations. Certain groups of patients respond to certain drugs and personalizing the drugs only for subpopulations that are responding to the disease reduces the failure rate in non-responding populations. Certain known drugs can be used for different indications, or newer combinations of known drugs can be tried. Overcoming the biologically unknown and simulating the disease process at the signaling and metabolic pathway level may be another method for decreasing the failure rate [21,22,25].

Systems biology finds various applications in drug discovery [21]. It helps to identify newer drug targets as gene expression data is incorporated in networks to find key nodes that can act as targets. This approach has led to the discovery of newer disease genes when known genes were validated [21]. The discovery of feedback loop mechanisms with cyclin dependent kinase 1 (CDK1) and wee1 like protein kinase (WEE1) or the discovery of microRNA (miRNA) target interactions in the development of lung cancer in non-smoking females, are examples of applications of systems biology [26]. 

However, many of these targets for drugs are not easily amenable to inhibition by small molecules. The systems biology tools can demystify the mechanism of actions of different compounds by understanding the process affected by the drug. Gene expression profiles reported from drug resistant and drug sensitive cell lines to Lapatinib in breast cancer patients pointed to the loss of ErbB2, a plasma membrane bound receptor tyrosine kinase pathway for glucose metabolism through network-scoring method, Netwalk [27,28,29].

Target based drug therapy, however, has not led to a substantial increase in the number of drugs entering the market. Phenotypic drug discovery, where drugs are tested in cellular or animal models, is experiencing a comeback. This is because the functional domain of the targets cannot be blocked by small molecules; the domains may be intracellular or the targets remain unknown, hence the renewed interest phenotypic drug discovery.

Cell system biology incorporates the principles of both phenotypic drug discovery and combination assays. BioMAP is a human cell-based assay that models disease in an in vitro format. It determines the efficacy, safety and mechanism of action of drug molecules on cell based models. [30]. The cells are selectively stimulated with pathway activators pertinent to cell signaling networks for the disease. The cultures, pathway activators/inhibitors and biomarker endpoints are selected on systems biology approach. The cell-signaling networks are more representative of in vivo conditions [30]. A diverse number of compounds can be detected on these assays, as they are highly interconnected and robust as cell signaling networks. Predictive models are built on these assays for pathway and target mechanisms derived from databases, such as REACTOME, Pathway Commons [31], KEGG, InnateDB [32], NCI-PID, and WikiPathways for in silico. 

## 3. Notch Signaling

A directed effort has been made to enhance our understanding of the signaling pathways like the Notch signaling system [33]. Notch genes were discovered and reported to induce a sex-linked mutation, which is expressed as serrated wings in *Drosophila melanogaster* [33]. It encodes transmembrane surface receptor interactions between notch receptor on the cell surface and a ligand on the other. Signal transduction affects the fate of the neighboring cell as the fate of one cell is affected [33]. Notch signaling is responsible for the fine-tuning of differentiation, proliferation and segregation of different lineages of cells. The core elements consist of notch receptors, DLL (δ-like), JAG (jagged) ligand and DNA binding proteins [34]. It is a highly intricate pathway where signaling is controlled due to various factors. First, Notch- and δ-receptors are extremely sensitive to gene dosage, which is associated with the intensity of the signal [33]. This may be due to the lack of enzymatic amplification as seen in some other pathways. Second, cis or trans signal interactions can give bidirectionality to the signal, hence the ratio of cis-trans interactions is an essential regulatory mechanism. Thus, any input which causes changes in the doses or changes in the cis-trans interactions can alter the expression of the gene. Thirdly, additional regulatory mechanisms like post-translational modifications of receptors or ligands can also alter the signaling mechanism [33]. 

In addition, signal interaction or cross talk between different pathways also governs morphogenesis or pathogenesis. These interactions take place in a more indirect manner, making it challenging to decipher at a molecular level. Some modifier screens can detect thousands of genes and are practically impossible to understand all the pathways and interactions between them. Hence genes, which interact in bona fide pathway components or can be recognized individually from more than one screen, should be prioritized. 

Notch signaling has a very complex genetic circuitry. A molecular hypothesis can be formed by finding out if these genes interacting with notch, affect the basic protein–protein interactions. Protein complex maps, like the RAS-MAPK signaling network, a communication process that controls the basic cellular activities, is available and modifier genes can be added to these maps as more information becomes available. *Drosophila* protein interaction map is the first-metazoan proteome physical map [35], which gives a holistic view of the interacting genes with Notch and how they relate to the various proteins within the cell. Such valuable insights obtained from these network maps are important in understanding the pathological role of Notch signaling and its effect in drug development. 

## 4. Systems Biology as a Biomarker

Systems biology is also used in biomarker activity to either assess the presence or severity of disease as well as the endpoint of treatment or stratification of patients in different disease classes that may alter the treatment (see Figure 2). Various chronic inflammatory diseases release numerous cytokines that interact to drive the pathogenesis of disease. The successful inhibition of tumor necrosis factor-α (TNF-α) in rheumatoid arthritis (RA) and various other chronic inflammatory diseases like Crohn’s disease proved that the communication of various cytokines was disrupted due to blockade of TNF-α. The systems biology approach has been utilized by Liu et al. [36] to study RA. After organizing a hierarchical clustering of the transcriptional profiles, they found approximately 244 set of genes that were differentially expressed between RA and normal individuals after performing a functional analysis. Moreover, 142 biological processes and 19 KEGG pathways were over-represented by these 244 genes. Another example is the central role of IL-17 in inflammatory bowel disease (IBD). Since, it has become more evident that manipulation of a specific cellular or microbial population in the gut may not lead to an IBD cure. It has become more vital to harness the potential of systems biology in order to develop a greater therapeutic potential. Polytarchou et al. [37] proposed to develop an IBD drug-molecular interactome map, based on the connectivity map platform. Balbas-Martinez et al. [38] demonstrated the success of a pharmacological model consisting of 43 nodes and 298 qualitative interactions in order to propose a therapy for IBD. Combined inhibition of cytokines [39], like inter-leukin (IL)-23 and IL-12, are also used in certain diseases. These are the shared vulnerable nodes and blockade of these nodes disrupts the disease. However, these nodes show hierarchy, e.g., blockade of IL-6 (though useful for treatment of rheumatoid arthritis) is not useful in other chronic inflammatory diseases. In another approach, Price et al. [40], calculated polygenic scores from genome-wide association studies (GWAS) for 127 traits and diseases, and used these to discover molecular correlates of polygenic risk and discovered that the genetic risk for inflammatory bowel disease was negatively correlated with plasma cystine. Mayer et al. [41] successfully demonstrated the success of using a small set of biomarkers identified by a systems biology approach to enhance the prediction of renal function loss in patients with type II diabetes. Clinical responsiveness to TNF-α is similar across the board and puts it high in the molecular taxonomy tree. Vectra DA, an omic blood multibiomarker, is another example of this approach [42]. Multibiomarker disease activity tested in rheumatoid arthritis patients affects the treatment decisions in a very early phase of the treatment. These expressions of gene characteristics that are derived from a data driven approach can be used as biomarker endpoint [42]. A pharmacodynamic biomarker can confirm if a drug is hitting the target and producing the desired effect and help in dose selection, as information for adverse effect if it is hitting an undesired target [43]. Flow cytometry approaches monitor multiple cell population and pathways, e.g., the cells and pathways of janus kinase (JAK) inhibitors in blood stimulated by various cytokines [42]. Anvar et al. [44] reported, via systems biology approach, the manifestation of hepatocyte nuclear factor 4 alpha (HNF4A), Transcription initiation factor subunit 1 (TAF1) () and Tumor suppressor gene P53 (TP53) as the most significant nodes in the interaction network of the engaged proteins in gastric cancer. Biomarkers can also help to assess patients who may respond and those who may not, thus improving the cost-benefit ratio or characterizing disease process and disease activity [43]. Piening et al. [45] studied integrated personal omics profiles during periods of weight gain and weight loss and reported that weight gain is associated with the activation of strong inflammatory and hypertrophic cardiomyopathy signatures in blood.

## 5. Summary

A systems biology approach requires an understanding of large and diverse data and its integration into meaningful physiology, pathway and disease processes to formulate newer treatments and effective drugs. Distinct cytokine inhibition has improved our ability to understand and overcome the molecular pathogenesis of various inflammatory diseases and their preclinical models. Various pharmaceutical companies and laboratories have pursued systems biology approaches on their own, but a unified approach is needed. Although the systems biology approach to drug discovery is in its initial growth phase, it will not be long before it can be used as a direct method for formulations of new drugs to treat a vast number of diseases in humans.

## Figures and Tables

**Figure 1 medsci-06-00043-f001:**
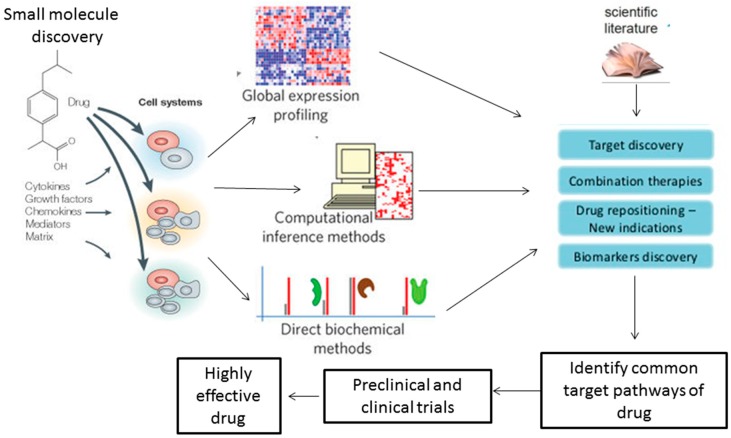
This scheme explains the role of systems biology in drug discovery, from the presence of a single chemical compound to the development of a highly effective drug.

**Figure 2 medsci-06-00043-f002:**
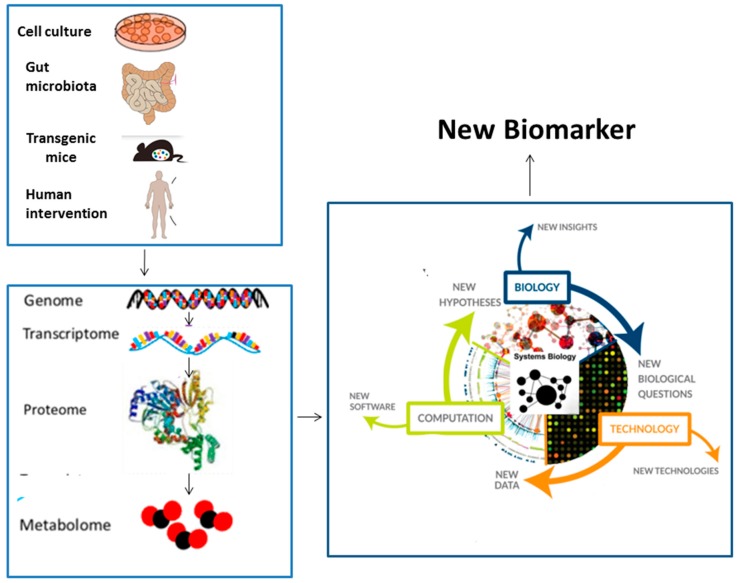
The role of systems biology in biomarkers, from the identification of a gene via human profiling towards systems biologic approaches in the identification of biomarkers.

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
