# Peer review of "The Role of Systems Biologic Approach in Cell Signaling and Drug Development Responses—A Mini Review"

_medsci, 2018, doi:10.3390/medsci6020043_

Reviewer 1 Report

1. There are many instances of the use of English being not quite correct, which may appear subtle to  non-native English speakers. Careful proof-reading by an native anglophone is required. A few examples:

"Recent studies in literature..."  have described its role should read "Recent reports in the literature....."

"However, the specific molecular mechanisms underlying these observations remains elusive" should read "However, the specific molecular mechanisms underlying these observations remain elusive"

Replace "In the ancient era" with "In the past"

etc

2. The review tends to describe generalities for the most part. It would benefit from a few more examples, as discussed for Notch signaling.

3. Section 4 adresses a very important aspect of disease, that of biomarkers.  It would help the reader if there are more details described in the text, as opposed to simply referring to another publications. As it stands one does not learn much from this section, without consulting the references cited, which means reading another 5 papers.

Author Response

Responses to the reviewer’s comments:

Thank you for your comments. We have taken them under serious consideration and have revised parts of the manuscript appropriately.

Reviewer # 1-

Comment #1: There are many instances of the use of English being not quite correct, which may appear subtle to  non-native English speakers. Careful proof-reading by a native Anglophone is required. A few examples:

"Recent studies in literature..."  have described its role should read "Recent reports in the literature....."

"However, the specific molecular mechanisms underlying these observations remains elusive" should read "However, the specific molecular mechanisms underlying these observations remain elusive"

Replace "In the ancient era" with "In the past"

Response to the comment: Thank you for the suggestion. We apologize for the vocabulary and grammatical errors. We have now corrected the manuscript with the help of an English language editing services. We have also had it reviewed by a native Anglophone.

2. The review tends to describe generalities for the most part. It would benefit from a few more examples, as discussed for Notch signaling.

Response to the comment:We apologize for the incomplete information. We have now added additional information in the manuscript.

3. Section 4 addresses a very important aspect of disease, that of biomarkers.  It would help the reader if there are more details described in the text, as opposed to simply referring to another publications. As it stands one does not learn much from this section, without consulting the references cited, which means reading another 5 papers.

Response to the comment:. We have now added additional information as well as cited the appropriate references.

Reviewer 2 Report

In the presented manuscript, Abhyankar et al. describe a summary of systems biologic approach and how useful it is for drug discovery and development and to comprehensively understand an intracellular signaling pathway. I have no major issues with the writing. However, it could be considered to change the title because the authors do not describe much about chemokine and cytokine response. Another criticism I would have is that there could be figure(s) added. That may help readers.

Author Response

Responses to the reviewer’s comments:

Thank you for your comments. We have taken them under serious consideration and have revised parts of the manuscript appropriately.

Reviewer #2:

Comment #1: In the presented manuscript, Abhyankar et al. describe a summary of systems biologic approach and how useful it is for drug discovery and development and to comprehensively understand an intracellular signaling pathway. I have no major issues with the writing. However, it could be considered to change the title because the authors do not describe much about chemokine and cytokine response. Another criticism I would have is that there could be figure(s) added. That may help readers.

 Response to the comment: Thank you for your positive comments. We have now changed the title to “ The role of systems biology in cell signaling and drug development”. Furthermore, we have added figures to the main text (See Figure 1 and 2).

Reviewer 3 Report

Authors have demonstrated the use of systems biology in medicine and drug discovery field with proper references. Generally the paper is well read and aimed for general readership. To improve this paper, it would be greatly appreciated if authors address well on my suggestions below.

Suggestions

1.     I recommend authors to include one of papers using weighted gene co-expression analysis (WGCNA) for the section mentioning "module repertoire analysis" (first paragraph of page 3), such as  Xue Z et al., Nature, 2013, PMID: 23892778. It is one of key research in terms of module analysis. 

2.     If authors need more review papers to cite in section 2, I can recommend a review paper about the use of systems biology in NASH/NAFLD treatment (Bosley J et al., Drug Discovery Today, 2017)

3.     In the last paragraph of page 4 (from line 149), literature that authors cited is very few. At least authors should consider citing resources of signaling pathways, such KEGG and REACTOME, or some papers like InnateDB (Karin Breuer et al., Nucleic Acids Research, 2013)

4.     Line 187 at page 5 should cite a reference. Does it refer to no. 29?

5.     Key part of this paper is section 4 and it doesn’t look much highlighted.  I recommend some key papers of wellness study to include, such as Nathan D Price et al., Nature Biotechnology, 2017 and Brian D Piening et al., Cell Systems, 2018.

Typos

6.     Line 127at page 4: biologic à biologically

7.     Line 147 at page 4: the sentence looks grammatically wrong.

Author Response

Responses to the reviewer’s comments:

Thank you for your comments. We have taken them under serious consideration and have revised parts of the manuscript appropriately.

Reviewer # 3:

Comment #1:     I recommend authors to include one of papers using weighted gene co-expression analysis (WGCNA) for the section mentioning "module repertoire analysis" (first paragraph of page 3), such as  Xue Z et al., Nature, 2013, PMID: 23892778. It is one of key research in terms of module analysis. 

Response to the comment: Thank you for the suggestion. We have now cited the paper in the main manuscript.

Comment #2:     If authors need more review papers to cite in section 2, I can recommend a review paper about the use of systems biology in NASH/NAFLD treatment (Bosley J et al., Drug Discovery Today, 2017)

Response to the comment: Thank you for the recommendation. We have now cited the above paper in section 2.

Comment #3:     In the last paragraph of page 4 (from line 149), literature that authors cited is very few. At least authors should consider citing resources of signaling pathways, such KEGG and REACTOME, or some papers like InnateDB (Karin Breuer et al., Nucleic Acids Research, 2013) 

Response to the comment: Thank you for the suggestion. We have now added the information to the manuscript. The sentence has been modified to the following , “Predictive models are built on these assays for pathway and target mechanisms derived from databases such as REACTOME, Pathway Commons[29], KEGG, InnateDB[30], NCI-PID and WikiPathways for insilico”.

Comment #4:     Line 187 at page 5 should cite a reference. Does it refer to no. 29?

Response to the comment: We apologize for the missing reference. We have now cited the reference in the manuscript, which is as you mentioned.

Comment #5:     Key part of this paper is section 4 and it doesn’t look much highlighted.  I recommend some key papers of wellness study to include, such as Nathan D Price et al., Nature Biotechnology, 2017 and Brian D Piening et al., Cell Systems, 2018.

Response to the comment: We apologize for the incomplete information. We have now added additional information as well as cited the appropriate references.